# Women's Participation in Firms' Management and Their Impact on Financial Performance: Pre-COVID-19 and COVID-19 Period Evidence

**Charalampos Basdekis [1], Ioannis Katsampoxakis [2,*] and Konstantinos Anathreptakis [3]**

1   Department of Tourism Management, School of Management, Hellenic Open University, 26335 Patras, Greece;
    basdekis.charalampos@ac.eap.gr
2   Department of Statistics and Actuarial–Financial Mathematics, School of Sciences, University of the Aegean,
    83200 Karlovasi, Greece
3   Alpha Bank, 10564 Athens, Greece; konstantinos.anathreptakis@alpha.gr
*   Correspondence: ikatsamp@aegean.gr

**Abstract:** At a time when gender equality is a key priority of all international organizations, this paper can be considered a remarkable contribution to the role of women executives in firms' performance. More specifically, this study focuses on the effect of women holding positions of responsibility on firms' performance worldwide. For the purposes of our research, we applied cross-sectional and panel data analysis for all sectors at an international level from 2019, the year preceding the breakout of the pandemic crisis, to 2021, while the indicators used to measure the participation of women in executive positions are classified as ESG indices. The empirical analysis findings end up showing that the participation of women in executive positions positively affects firms' performance over time, while there is no material change observed before and during the COVID-19 pandemic period. More specifically, when the percent of women processing job positions of responsibility increases by 10%, then the index of profitability will increase from 1.4% to 1.8%, regardless of the measurement of female participation in executive positions used. The results of this study constitute a remarkable contribution to the promotion of the creative economy, the progress of societies, and sustainable development. The research's outcome can be primarily used by policymakers drawing up policies for achieving gender equality in the labor market and workplaces and by shareholders and firms' managers in order to trust females in executive positions in favor of their firms' financial performance. The current study is unique in that it focuses on the period before and during the COVID-19 period, as a period of high volatility in economic activity worldwide, while the sample includes firms from large and mid-cap companies belonging to developed and emerging markets. The above approach will contribute to providing more credible information related to the role of women executives in firms' performance.

**Keywords:** women on board; female executives; return on equity; return on assets; ESG criteria

## 1. Introduction

The basis of the creative economy is capturing the creativity and imagination of individuals in order to upgrade and reinforce an idea and add value. The term creative economy has evolved over time and refers to all economic activity. The creative economy is activated when human beings create the appropriate conditions, so that creativity can be considered the main spring of value added, which mostly affects any kind of transaction [1,2]. Creative economy emphasizes, as expected, creativity and defines it as the steam engine of technological advance and innovation, thus leading companies to track down, retrieve, and reclaim business opportunities and acquire strategic advantages. Therefore, it will be very interesting and essential for a sustainable economy to highlight the importance and leadership role of women in firms' performance.

According to [3], the active participation of females in the creative economy can be framed in terms of gender equality, which signals the progress of societies and is the key to sustainable development.

Gender equality is nowadays considered a deterministic factor in societies' evolution. The [4] has determined gender equality as the basic priority related to their strategy of society's evolution and the economy's development and included it as a top issue that should be implemented according to Agenda 2030, through which the appropriate conditions for sustainable development will be created.

If policies of equal treatment between genders are not immediately implemented both in society and the workplace, there is a risk of undermining economic stability worldwide. Investing in female empowerment creates a multiplier impact on families, societies, and economies. This in turn will lead to society's reform, and everybody will be able to perceive females' potential and value in society and the labor market.

According to [5], the workforce rate was decreasing for both men and women worldwide, even before 2020 and the breakout of the COVID-19 health crisis. However, the reduced participation of females in the labor market as compared to males is an intense global issue. This phenomenon is unfortunately verified by the fact that despite the increased rate of female enrollment in tertiary education, there is still, globally, a remarkable gap between males and females in specified work positions, to the detriment of women.

Thus, as it is apparent, the concept of gender equality has to lead to an immediate and fair reallocation of resources, responsibilities, opportunities, and job positions between genders without displeasing any gender. The elimination of discrimination between genders will lead to gender equality and the acquisition of equal rights [6].

Moreover, social norms and institutions differ across countries, leading to discrepancies in the way of treating males and females in the labor market [7]. The situation of gender differentiation worsens in cases involving religious issues [8].

This kind of equality is therefore just as important in the case of entrepreneurial activity and social and economic sustainability. After all, gender equality in executive job positions and participation in boards of directors contribute to the upgrading of morality and the change of culture and mentality in societies and is the basis for creating more fair societies that will not violate human rights and respect human diversity. In such a context, the European Union embraces the principles set by the United Nations and has published directives related to the reform of the labor market, imposing on European firms' specific minimum percentages of women in job positions and facilitating the entrance of women in executive positions and on boards of directors. Despite this intense and coordinated effort, there is still a long way to go until the ultimate goal of gender equality in the labor market is achieved [9].

In such a context, it will be very important to investigate whether females play a leading role in entrepreneurial development, sustainability, and firms' profitability. It has to be noted that institutions and governments have globally promoted women's active participation in firms' boards. Our research interest focuses on the influence of female leaders on firms' performance worldwide. More specifically, in addition to the importance and role of females in firms' profitability, this study uses even more factors that can be considered of major importance in determining firms' profitability. These additional factors are sales growth and financial leverage. For the purposes of our paper, profitability is measured by both ROE and ROA as proxies of profitability in order to clarify whether the way of measuring profitability can differentiate the existence and extent of any influence of factors implied on profitability. It should also be stressed that our study's findings highlight the 'bright side' of women in governance, suggesting that the presence of women in double leadership positions can amplify the benefits related to firms' performance, contributing decisively to their competitiveness in an ever-intensifying business environment.

The participation of women in the Board of Directors and as executives both meets ESG criteria and verifies their positive role in companies' performance worldwide. This is useful for stakeholders, and obviously, it sheds more light on academic research. More specifically,

our outcome could be used by international organizations struggling for human rights and gender equality both in society and more specifically in the labor market, in order to draw up policies of equality and assert human rights in workplaces. Moreover, shareholders who perceive the positive impact of female executives on their companies' performance will be able to make more wise and profitable decisions regarding the allocation of workforce in job positions of responsibility, looking ahead to the future performance of their companies.

The United Nations' main goal of sustainable development considers that females' empowerment is of major importance, as they perceive that the advantages of female nature and their competences, which in most cases differ from those of males, are not utilized in the labor market, which negatively affects societies' evolution and companies' performance.

Moreover, in the last few years, specific EU directives were published that focus on the elimination of each kind of gender discrimination in working places for all companies performing within the EU. Thus, taking into consideration the United Nations' and EU's approaches, we consider that this study will prove to be of major importance and value added for the role of females in firms' performance, creating the necessary conditions and providing the appropriate incentives for the new generation of women to struggle and assert professional respect worldwide.

As far as the structure of the paper is concerned, it firstly includes the most important developments in the international literature for the issue under consideration. This part is divided into three sub-parts for better assimilation and analyses issues such as the impact of the gender gap on wages and the effect of firms' size and leverage and female executives on firms' profitability. The next section focuses on the research hypothesis set and includes the methodological part, where the sample and the model under examination are analyzed. In Section 4, the results of the empirical approach are imprinted and analyzed, while the paper ends up with the gleaning of the most important outcomes, research limitations, policy implications, and propositions for further research.

## 2. Literature Review

### 2.1. The Impact of the Gender Gap on Wages

A highly significant issue attributable to gender discrimination in workplaces is the gender wage gap. A very representative fact of the current situation is that the wage discrimination gap globally is expected to close no earlier than before the next 250 years [10]. The issue of institutional strength can contribute to a more thorough understanding of gender wage discrimination across countries [7]. This settled wage regime exists as well as in the case of the transition from male to female executive officers and vice versa [11]. Thus, taking into consideration these parameters, gender wage discrimination appears to have even more intense features, although women CEOs are a minority at an international level, regardless of the economic situation, background, or potential of the economy [12].

However, academic studies lead to contradictory or mixed results, which mainly depend on the country's sample specification [13]. The most dominant research view is that male executive officials are better compensated related to females holding positions of responsibility [14,15], focusing on the existence and spread of stereotypes related to the ability of females to acquire and indicate leadership competences. However, [16] fail to find evidence of gender CEO wage disparities in Sweden, as well as [17,18] for the UK. In the same direction are the findings of [19,20], who focused their studies on CEO-specific features. The differentiation of gender wage disparities related to executives between countries is mainly dependent on a culture oriented to the gender equation and the legal framework, which in some cases (i.e., the USA, UK, and Scandinavian countries) forbid the implementation of such disparities [16,17,21].

### 2.2. Firms' Size, Leverage, and Profitability

Profitability can be measured through different ways. The proxies that are most commonly used in the international literature are Return on Equity (ROE) and Return on Assets (ROA). ROE is the ratio of net income to shareholders' equity capital and indicates

the degree to which firms increase their profitability in relation to the invested capital [22]. On the other hand, ROA is calculated through net income by total assets and determining whether a firm has the ability to use its assets in an efficient manner to generate profits.

In the international literature, there are plenty of studies focusing on factors affecting firms' profitability. As it has already been stated, there are many factors affecting, each in their own way, firms' profitability, while in most cases the results seem contradictory.

However, there is still enough space for studying the impact of gender equality and the role of female executives on firms' profitability. It should not be taken just as a coincidence that there is a high gender gap in the labor market between highly educated women and men worldwide. This is the case in our research, as highly educated persons are selected to cover job positions of high responsibility.

More specifically, firm size affects profitability in a positive way. This can be explained by the fact that large companies have the ability to obtain financing from different sources at lower costs. So, these companies have easier access to global money and capital markets, enjoying a more competitive cost of capital [23,24]. This positive relationship is confirmed by [25] research, using sales' growth as a proxy for firms' size. On the other hand, the arduousness of small firms to obtain easy and cheap financing may affect their profitability and can manipulate different accounting situations to indicate higher profitability [26,27].

Another considerable factor affecting firms' profitability is financial leverage. Financial leverage is directly linked to debt increases and contributes to returns' volatility increasing. According to a literature review, there is a negative correlation between profitability and firms' leverage [28,29]. At the same bandwidth, ref. [30] found that there is a negative influence of the debt-to-equity ratio and their country's yield bond on firms' profitability, regardless of the proxy used, while there is a positive impact of firms' size and the ECB's monetary policy on firms' profitability. Moreover, levered firms seem to generate more profits than unlevered ones [31], while the increased profitability of levered firms contributes to the quicker repayment of their debt.

According to [32], firms' size and financial leverage may lead to ambiguous outcomes. This is due to firms' liquidity, which affects sales, investments, and the need for external financing. Ref. [33] found a negative relationship between both firms' size and financial leverage on their profitability, while [34] concluded that firms' size affects profitability in a negative way [35] and financial leverage in a positive one. Refs. [36,37] concluded that firm size and leverage indicate a positive impact on firms' profitability up to a certain turning point after which the impact becomes counterproductive.

### 2.3. Female Executives and Firms' Performance

Our research interest leads us to seek answers to questions related to the role of women in executive positions or on corporate boards worldwide. Different cultural backgrounds, different mentalities, different experiences, different opinions, and different legal systems and laws implementation play a significant role in perceiving and encountering specific social issues with economic extension, such as discrimination in workplaces. Moreover, the participation of women on firms' boards is part of the Environmental, Social, and Governance (ESG) group of fields.

More specifically, in the European Union, there are some specific directives related to the issue of equal treatment in the labor market. Firms that operate in EU countries are obliged to implement and harmonize the human resource strategy according to European Parliament Directive 2002/73/EC and Council Directive 76/207/EC. According to these directives, it is required that firms treat both male and female genders equally in terms of conditions at working places, preferment, employment, education, and job training. These directives inspired us to test the importance, role, and participation of female executives in helping firms perform effectively.

The studies being conducted concluded with contradictory results, as there are studies that end up with positive statistical significance between females' participation in boards of directors, firm effectiveness, and their financial returns [5,38–41]. On the contrary,

there are studies concluding that female executives have a negative impact on firms' performance [9,13,42]. The above studies results may differ according to sector, geographic area, and the firms' conditions and influences derived either from the internal or external environment. In such a case, the net gender executive result on financial performance could be either positive, negative, or even neutral [38]. Moreover, it should be mentioned that one more crucial issue, supplementary to the quality and competences of women in participating in the boards of directors, is the mass of females, as there is a critical mass of at least three women executives that can affect substantially board performance and upgrade the innovativeness of companies [43].

However, most of the most recent studies seem to end up with convergent outcomes related to the role of females holding positions of high responsibility and their impact on firms' performance, regardless of the region under consideration and the active role of women in their societies.

More specifically, according to [44] findings, board size, audit committee independence, and audit committee meetings indicate a significant impact on the financial performance of Saudi Arabian listed firms, while board size and the general experience of top management seem to have a negative association with corporate performance. Ref. [45] highlighted the essential role of female directors on boards of companies in sub-Saharan Africa, proving that they affect corporate financial performance enhancement. Ref. [46] findings indicate that the presence of women in corporate governance in Pakistan-listed companies is positively associated with firms' financial performance.

Moving ahead to the role of women in key positions in more advanced Western economies, the conclusions seem to amplify both the conducted analysis and the international literature. Ref. [47] focused their analysis on Standard & Poor's 500 companies belonging to the information technology sector, and they concluded that there is a positive influence of women participating in corporate boards on companies' performance. However, this is not verified in the case of the percentage of female executives and their impact on firms' performance using ROA as a proxy variable. Ref. [48] showed that women's participation in boards in European countries can enhance the financial performance of a company and is moderated by many cultural factors. Moreover, ref. [49], from their point of interest, found that CEO duality has a positive effect on corporate performance when a woman holds both the roles of CEO and board chair in firms performing in European countries. Another important study that shows the role of the attitude of "strength in unity" is that of [50], who found that when firms have a top female manager and ownership is exclusively male, firms show higher average labor productivity. However, these results are very heterogeneous among regions.

## 3. Hypothesis, Data and Methodology

### 3.1. Hypothesis Testing

Within the entire international literature framework and the efforts put forth by international organizations, there has arisen a reasonable question regarding whether and in what way the participation of women in corporate decisions affects firms' financial performance worldwide. Participation in corporate decisions can be approached both by the participation of women on the boards of directors and by the participation of women in executive positions in companies' organizational charts. This question becomes even more important as it includes the pre-COVID-19 period and the epicenter of the pandemic period. Thus, it would be of great interest to examine whether there have been any material differences observed through the years under consideration. This study could provide interesting new findings to the existing literature as it differentiates the analysis by taking into consideration all economic activity sectors worldwide before and during the pandemic period and applying different proxies for measuring women's impact on firms' performance.

### 3.2. Data and Methodology

The sample of our study is based on the member companies of the Bloomberg World Equity index, called the Bloomberg World Large and Mid Cap Index, which is a float market-cap-weighted equity benchmark that covers 85% of the market cap of the measured market at the end of March 2023. This index consists of listed securities that provide investors with the building blocks for investing in a range of markets and size segments across the globe in a consistent, rules-based manner while maintaining a balance between broad market coverage and liquidity. Moreover, the Bloomberg Global Equity Index offers various size and regional segmentations and an array of services in investable index products. More specifically, this index includes developed markets and emerging markets, large and mid-cap companies, and is considered a broad and diversified indicator in order to be more reliable in the outcome of the analysis.

The total number of companies in our sample is 3332 from 58 countries, and the data used for the purposes of our analysis are on a yearly basis, so the total observations of all years of our sample are 9996. The sample includes the years 2019, 2020, and 2021. The reason for choosing the period extending from 2019 to 2021 for the purposes of our study is related to the purpose of our research, which is to examine the impact of women's participation in decision-making positions during the period of the pandemic crisis, starting just before its breakout.

This study examines the impact of women acquiring positions of responsibility and specific firms' factors on corporate profitability. For the purposes of our analysis, we apply two different measures of profitability (ROE and ROA) and two indices of women's participation in taking corporate decisions (percentage of women on board and female executives) in order to obtain a more accurate picture of the situation being shaped.

The model under consideration for the purposes of our analysis is the following:

$$Y_{t,i} = a_0 + a_1 X_{t,i} + a_2 Sales\_Growth_{t,i} + a_3 Total\_Debts\_to\_Total\_Assetss_{t,i} \tag{1}$$

More specifically, $Y_{t,i}$ corresponds to the dependent variable for firm *i*, at time *t*, and is divided into two variables, $ROE_{t,i}$ and $ROA_{t,i}$, for firm *i*, at time *t*, respectively, according to the model's specifications. According to the model's specifications, the $X_{t,i}$ variable is distinguished into two variables, percentage of women on board and the percentage of female executives. Thus, as it can be perceived, there are four model specifications that will be applied and tested.

The model specifications are be summarized as follows:

$$ROE_{t,i} = a_0 + a_1 WoB_{t,i} + a_2 Sales\_Growth_{t,i} + a_3 Total\_Debts\_to\_Total\_Assetss_{t,i} \tag{1a}$$

$$ROE_{t,i} = a_0 + a_1 FE_{t,i} + a_2 Sales\_Growth_{t,i} + a_3 Total\_Debts\_to\_Total\_Assetss_{t,i} \tag{1b}$$

$$ROA_{t,i} = a_0 + a_1 WoB_{t,i} + a_2 Sales\_Growth_{t,i} + a_3 Total\_Debts\_to\_Total\_Assetss_{t,i} \tag{1c}$$

$$ROA_{t,i} = a_0 + a_1 FE_{t,i} + a_2 Sales\_Growth_{t,i} + a_3 Total\_Debts\_to\_Total\_Assetss_{t,i} \tag{1d}$$

where:

Return on Equity (ROE): $ROE_{t,i}$ for firm *i*, at time *t*, as indicated, is a dependent variable and corresponds to an indicator that measures firms' profitability on a yearly basis. This index reveals the level of profits a firm earns, taking into consideration the invested funds of shareholders. Alternatively, ROE, as a measure of financial performance, is calculated by dividing net income by shareholders' equity. Taking into consideration that shareholders' equity is equal to a firm's assets minus its debt, ROE is considered the return on net assets.

Return on Assets (ROA): $ROA_{t,i}$ for firm $i$, at time $t$, as aforementioned, is a dependent variable and corresponds to an indicator that shows how profitable a company is relative to its total assets for each year. Return on Assets gives a clear picture of how efficiently a company manages its assets to generate earnings. More specifically, ROA is usually calculated by dividing a firm's net income by its average total assets. The average of a firm's total assets is calculated by adding the prior period's ending total assets to the current period's ending total assets and dividing the result by two.

Percent of women on board (WoB): $WoB_{t,i}$ for firm $i$, at time $t$, measures the percentage of women participating on the board of directors (BoD). This indicator is available on a yearly basis and is sourced solely from the company's primary corporate governance filing. It is impressive that there has not been any reasonable progress towards increasing the proportion of women on boards. In a global level, women occupy just 20% of board seats and continue to be excluded from the highest levels of corporate leadership [51]. It is noted that this indicator is essential for the design and creation of the ESG indicator of gender equality.

Percent of female executives (FE): $FE_{t,i}$ for firm $i$, at time $t$, measures the percentage of companies' female executives or female members of an equivalent management/executive body. Executives are the members of firms' workforce who possess job positions of high responsibility. This indicator is available on a yearly basis and is sourced solely from the company's primary corporate governance filing. According to [51] even less, compared to the percentage of women on board, is the percentage of women being occupied as executives from a Chief Executive Officer (CEO) perspective (5.0% globally). This indicator is essential for the design and creation of the ESG indicator of gender equality. It has to be mentioned that the higher the percentage of women on the board of directors and that of female executives, the higher the expected governance score of each company [52].

Sales growth: $Sales\_Growth_{t,i}$ for firm $i$, at time $t$ is an index that imprints the growth of revenue on a yearly basis. A percentage change in sales revenue is considered by comparing the current fiscal year vs. the previous one. Comparing revenue between two accounting periods demonstrates the firm's rate of growth, either positive or negative. It should be noted that revenue growth cannot be computed if revenue changes from the prior year to the current year.

Total Debt to total Assets: $Total\_Debts\_to\_Total\_Assetss_{t,i}$ for firm $i$ at time $t$. The ratio is used as a leverage ratio in percentage and defines the total amount of debt relative to total assets every year. The current ratio is used to show how much of a firm's percentage is owned by its creditors compared with how much of the firm's assets are owned by its shareholders. It has to be noted that the use of this indicator enables leveraged comparisons across different companies.

All models are applied in cross-sectional analysis for each year separately (2019, 2020, 2021) and panel data analysis for all years and firms simultaneously in the sample, respectively. Moreover, all of the specifications of the model were applied to all samples, which include firms from all activity sectors worldwide, and to all samples excluding financial sector companies due to their specifications in creating financial statements.

It should be noted that our sample consists of more cross-sectional values (3332 companies) compared to time series variables (three years), so the issue of heteroskedasticity seems to arise in our analysis. For cross-sectional analysis, in order to deal with heteroskedasticity, we used the Huber–White covariance method, so standard errors are heteroskedasticity-consistent [53,54]. Applying this method, it is possible to achieve coefficient covariance estimators that are robust to the presence of heteroskedasticity. The Huber–White robust standard errors are equal to the square root of the elements on the diagonal of the covariance matrix. White (1980) [54] derived a heteroskedasticity-consistent covariance matrix estimator that provides consistent estimates of the coefficient covariances in the presence of conditional heteroskedasticity of unknown form.

For panel data analysis, we calculated heteroskedasticity and serial correlation consistent standard errors within groups estimates [55]. So, we applied the White cross-section

coefficient covariance method to deal with cross-section heteroskedasticity. The White cross-section method assumes that the errors are cross-sectionally correlated. The method treats the pool regression as a multivariate regression and computes robust standard errors for the system of equations. This estimator is robust to cross-equation (contemporaneous) correlation and heteroskedasticity.

## 4. Empirical Analysis

### 4.1. Descriptive Statistics

Table 1 presents the descriptive statistics of the core variables used to test the influence of women's participation, either as executives or on boards of directors, on firms' profitability.

**Table 1.** Descriptive statistics.

| Percent of Female Executives | | | | Percent of Women on Board | | | |
|---|---|---|---|---|---|---|---|
| Year | Mean | Std. Dev. | Obs. | Year | Mean | Std. Dev. | Obs. |
| 2019 | 14.09 | 14.53 | 3063 | 2019 | 19.18 | 13.95 | 3110 |
| 2020 | 14.92 | 15.11 | 3170 | 2020 | 20.71 | 14.05 | 3209 |
| 2021 | 16.31 | 15.88 | 3180 | 2021 | 22.22 | 14.25 | 3220 |
| All | 15.12 | 15.21 | 9413 | All | 20.72 | 14.14 | 9539 |

More specifically, the percent of female executives and the percent of women on board are presented, regardless of the geographical area in which firms operate and the activity sector. As it can be observed, both the percentage of female executives and the participation of women in the Board of Directors (BoD) increased considerably in 2021 compared to 2019 and 2020, respectively. These results indicate an improvement in the climate in favor of the role and value of women in business activity, as the average percent of female executives and percent of women participating in boards have increased more than 2% in 2021 compared to 2019. These results verify that despite the problems and obstacles that have to be overcome, international institutions endeavors to achieve a more balanced and fair society and labor market seem fertile. It has to be noted that firms worldwide faced multiple issues, including the pandemic crisis, which slowed down their activities intensively and abruptly. However, it is an encouraging step for gender equality that firms have invested in the upgrading of women's roles in workplaces.

### 4.2. Regression Models

The key question seeking an answer is whether women's participation in corporate decisions can lead to profitable outcomes. We used two proxies for women's participation: the first is the percent of women participating on the board of directors, and the second is the percent of female executives. It should also be noted that in all cases of empirical analysis, there are applied and other considerable variables that affect firms' performance. These variables affecting firms' performance are sales' growth, which is directly linked with economic activity, and leverage, which is related to firms' financing.

The empirical analysis of this paper is divided into sub-empirical sections as cross-sectional and panel data analyses are conducted. Cross-sectional analysis (Tables 2 and 3) tests for the impact of women participating in BoD (Table 2) and female executives (Table 3), respectively, on firms' profitability for each year separately from 2019 to 2021. More specifically, Table 2 (a) and Table 3 (a) focus on a cross-sectional analysis of the impact of women participating in boards of directors and female executives, respectively, on all firms', including the sample's, performance for each of the years under consideration. On the other hand, Table 2 (b) and Table 3 (b) summarize the results of the influence of the corresponding variables on corporate profitability, excluding financial sector firms, for the period 2019–2021. So far, for the purposes of our empirical analysis, this segmentation of our sample, while applying it to the specific regression models, is due to the different balance sheet structures of financial companies compared to the others.

**Table 2.** (a) Cross-sectional regression analysis with women on the BoD as an independent variable (all firms). (b) Cross-sectional regression analysis with women on the BoD as an independent variable (excluding financial sector firms).

| | (a) | | | | | |
|---|---|---|---|---|---|---|
| **Year** | **2019** | **2019** | **2020** | **2020** | **2021** | **2021** |
| Dependent Variable | ROE | ROA | ROE | ROA | ROE | ROA |
| WoB | 0.1766 *** | 0.0304 ** | 0.1255 *** | 0.0130 | 0.1277 *** | 0.0278 ** |
| t-Statistic | 3.9081 | 2.1084 | 3.4123 | 0.6774 | 3.7865 | 2.0218 |
| Sales Growth | −0.0005 *** | −0.0003 *** | 0.0003 *** | 0.0003 *** | 0.0001 | 0.0001 *** |
| t-Statistic | −4.9761 | −4.8768 | 90.7280 | 67.1841 | 1.3556 | 2.9697 |
| Total Debt to Total Assets | 0.0601 | −0.0698 *** | −0.0980 ** | −0.1523 ** | −0.0201 | −0.0839 *** |
| t-Statistic | 1.3172 | −3.8603 | −2.3989 | −2.5135 | −0.512 | −4.4806 |
| C | 10.6588 *** | 7.0206 *** | 12.3030 *** | 8.4784 *** | 14.3868 *** | 7.8882 *** |
| t-Statistic R-squared | 9.2213 0.0109 | 13.8442 0.0270 | 10.5169 0.0126 | 6.5488 0.1112 | 12.0928 0.0037 | 13.6764 0.0208 |
| Adjusted R-squared | 0.0099 | 0.0261 | 0.0117 | 0.1104 | 0.0028 | 0.0199 |
| F-statistic | 11.0891 | 28.5939 | 13.2291 | 132.9339 | 3.8857 | 22.6211 |
| Prob(F-statistic) | 0.0000 | 0.0000 | 0.0000 | 0.0000 | 0.0087 | 0.0000 |
| Included observations | 3024 | 3090 | 3104 | 3191 | 3125 | 3201 |
| Huber–White covariance method | YES | YES | YES | YES | YES | YES |
| | (b) | | | | | |
| **Year** | **2019** | **2019** | **2020** | **2020** | **2021** | **2021** |
| Dependent Variable | ROE | ROA | ROE | ROA | ROE | ROA |
| WoB | 0.1925 *** | 0.0434 ** | 0.1554 *** | 0.0288 | 0.1423 *** | 0.0379 ** |
| t-Statistic | 3.5993 | 2.5291 | 3.6381 | 1.0853 | 3.5368 | 2.2883 |
| Sales Growth | −0.0005 *** | −0.0003 *** | 0.0003 *** | 0.0003 *** | 0.0001 | 0.0001 *** |
| t-Statistic | −4.7514 | −4.9990 | 89.6184 | 55.3388 | 1.2860 | 2.9846 |
| Total Debt to Total Assets | 0.0559 | −0.0917 *** | −0.1474 *** | −0.1873 *** | −0.0442 | −0.1113 *** |
| t-Statistic | 0.9798 | −3.9621 | −2.9897 | −2.6455 | −0.7976 | −4.5845 |
| C | 10.6639 *** | 8.0291 *** | 13.2717 *** | 9.7330 *** | 15.1194 *** | 9.1499 *** |
| t-Statistic R-squared | 7.8288 0.0120 | 12.9709 0.0398 | 9.7356 0.0170 | 6.5359 0.1404 | 10.6193 0.0041 | 12.9818 0.0308 |
| Adjusted R-squared | 0.0108 | 0.0387 | 0.0158 | 0.1394 | 0.0030 | 0.0297 |
| F-statistic | 10.1651 | 35.6564 | 14.8833 | 145.3129 | 3.5918 | 28.3743 |
| Prob(F-statistic) | 0.0000 | 0.0000 | 0.0000 | 0.0000 | 0.0131 | 0.0000 |
| Included observations | 2521 | 2585 | 2588 | 2673 | 2612 | 2686 |
| Huber–White covariance method | YES | YES | YES | YES | YES | YES |

*** 1% statistic significant, ** 5% statistic significant.

Applying cross-sectional regressions, it can be observed that almost in all cases examined, there is a statistically significant impact of the variables examined on firms' profitability. More specifically, the participation of women on the board of directors tends to affect the firm's profitability positively and intensively for all years examined, regardless of the proxy for profitability used. The only case in which there is no observed statistical significance at the 5% significance level is when ROA is applied as a proxy of profitability for 2020 (Table 2).

**Table 3.** (a) Cross-sectional regression analysis with female executives as an independent variable (all firms). (b) Cross-sectional regression analysis with female executives as an independent variable (excluding financial sector firms).

| | **(a)** | | | | | |
|---|---|---|---|---|---|---|
| **Year** | **2019** | **2019** | **2020** | **2020** | **2021** | **2021** |
| Dependent Variable | ROE | ROA | ROE | ROA | ROE | ROA |
| FE | 0.1654 *** | 0.0237 | 0.1475 *** | 0.0377 *** | 0.1609 *** | 0.0415 *** |
| t-Statistic | 3.3726 | 1.4469 | 3.3121 | 2.8007 | 4.4059 | 2.6184 |
| Sales Growth | −0.0005 *** | −0.0003 *** | 0.0003 *** | 0.0003 *** | 0.0001 | 0.0001 *** |
| t-Statistic | −5.7937 | −5.0576 | 40.2645 | 47.8302 | 1.5756 | 3.1605 |
| Total Debt to Total Assets | 0.0674 | −0.0665 *** | −0.0932 ** | −0.1517 ** | −0.0198 | −0.0834 *** |
| t-Statistic | 1.4524 | −3.6467 | −2.2126 | −2.5038 | −0.4371 | −4.4042 |
| C | 11.5264 *** | 7.1510 *** | 12.5643 *** | 8.1604 *** | 14.6144 *** | 7.8188 *** |
| t-Statistic R-squared | 10.3453 0.0105 | 14.3371 0.0250 | 11.9804 0.0148 | 5.3884 0.1133 | 13.6255 0.0072 | 14.6511 0.0227 |
| Adjusted R-squared | 0.0095 | 0.0240 | 0.0139 | 0.1124 | 0.0063 | 0.0218 |
| F-statistic | 10.5469 | 25.9620 | 15.3686 | 134.0887 | 7.4939 | 24.4653 |
| Prob(F-statistic) | 0.0000 | 0.0000 | 0.0000 | 0.0000 | 0.0001 | 0.0000 |
| Included observations | 2979 | 3044 | 3066 | 3153 | 3086 | 3162 |
| Huber–White covariance method | YES | YES | YES | YES | YES | YES |
| | **(b)** | | | | | |
| **Year** | **2019** | **2019** | **2020** | **2020** | **2021** | **2021** |
| Dependent Variable | ROE | ROA | ROE | ROA | ROE | ROA |
| FE | 0.1851 *** | 0.0402 ** | 0.1726 *** | 0.0582 *** | 0.1843 *** | 0.0570 *** |
| t-Statistic | 3.612 | 2.0789 | 3.4429 | 3.4737 | 4.2655 | 3.0223 |
| Sales Growth | −0.0006 *** | −0.0003 *** | 0.0003 *** | 0.0003 *** | 0.0001 | 0.0001 *** |
| t-Statistic | −5.5882 | −5.2772 | 34.1882 | 36.1916 | 1.5689 | 3.2903 |
| Total Debt to Total Assets | 0.0665 | −0.0876 *** | −0.1402 *** | −0.1872 *** | −0.0446 | −0.1114 *** |
| t-Statistic | 1.1682 | −3.7497 | −2.7250 | −2.678 | −0.778 | −4.5260 |
| C | 11.4857 *** | 8.1636 *** | 13.7472 *** | 9.4818 *** | 15.3684 *** | 9.0994 *** |
| t-Statistic R-squared | 8.48034 0.0121 | 13.0440 0.0380 | 11.1068 0.0195 | 5.2908 0.1450 | 11.5499 0.0085 | 13.3714 0.0343 |
| Adjusted R-squared | 0.0109 | 0.0368 | 0.0183 | 0.1440 | 0.0073 | 0.0332 |
| F-statistic | 10.0676 | 33.3819 | 16.8804 | 148.7387 | 7.3110 | 31.2953 |
| Prob(F-statistic) | 0.0000 | 0.0000 | 0.0000 | 0.0000 | 0.0001 | 0.0000 |
| Included observations | 2477 | 2541 | 2551 | 2636 | 2574 | 2648 |
| Huber–White covariance method | YES | YES | YES | YES | YES | YES |

*** 1% statistic significant, ** 5% statistic significant.

As far as the impact of sales growth is concerned, the output provides a picture of a very low effect of sales' growth on firms' profitability. More specifically, for 2019, there

is a very weak negative impact of sales growth on firms' profitability, regardless of the profitability proxy used. This situation does not seem to change for years 2020 and 2021, regardless of the profitability index under examination, as there is a faintly positive effect. Thus, the results extracted do not provide us with a clear and robust picture related to the impact of sales' growth on firms' financial performance, either positive or negative (Table 2).

Moreover, the empirical results of Table 2 indicate the existence of a negative relationship between leverage and firms' profitability over the years, except for the impact at the 5% statistical significance level when applying ROE as a profitability proxy. This could be the case when the firms are overdebted [34,56,57]. However, due to the pandemic, the situation during the years tested was weird. The outbreak of the COVID-19 pandemic during 2020 and 2021 affected firms' activities substantially and negatively, so this could decrease the total assets to total debt ratio.

It is remarkable to point out that the outcome of the cross-sectional regression analysis, applying the percentage of women executives as the main deterministic variable, is almost identical with that of the previous case (women participating in boards of directors). More specifically, it can be observed that female executives affect in a positive, statistically significant way firms' profitability for all years examined, regardless of the performance index applied, while there does not seem to exist any other discrepancy in the way and significance of factors affecting firms' profitability (Table 3).

As can be derived from the above analysis, there are no material differences in our model when we excluded the companies in the financial sector compared to all of the sample companies. Moreover, the statistically significant results in 2019, 2020, and 2021 verify the stability of the approach, as the extracted results are not differentiated either related to the time period and the sample under consideration or the variable used to determine women's role in businesses. The period under consideration is characterized by high volatility as it encompasses the impact of the 2020 pandemic and the strong recovery of 2021. However, despite the high volatility in corporate data, the empirical results of our models were very stable.

Following our analysis, panel data regression analysis is proceeding. Table 4 summarizes the results of panel data analysis. Table 4 (a) includes the results of all the sample's firms, while in Table 4 (b), the financial sector's firms were excluded from the analysis. Panel data regression analysis is very important for the empirical analysis of our study and provides a more accurate picture, as it uses data from most activity sectors, over many points in time simultaneously. It is very interesting to note that the results obtained are very similar to the outcomes derived from cross-sectional analysis.

There is a positive and significant impact of both female participants in BoD and female executives on firms' performance, regardless of the profitability proxy used. This result is in line with that of the cross-sectional analysis. Our results are moving in the direction of other studies' results [9,39–41,45,47,58].

The same outcome is observed in the case of sales growth's impact on firms' profitability, as derived from the cross-sectional analysis for the last two years of the sample (2020, 2021). These results are also verified in the scientific works of [23–25,32]. As far as financial leverage is concerned, there is a negative statistically significant relationship at the 5% significance level only while applying ROA as a profitability proxy variable, as occurring, and in cross-sectional analysis (Table 4). These results are in accordance with the studies of [30,33,36,37].

**Table 4.** (a) Panel regression analysis (all firms). (b) Panel regression analysis (excluding financial firms).

| (a) | | | | | |
|---|---|---|---|---|---|
| **Dependent Variable** | **ROE** | **ROA** | **Dependent Variable** | **ROE** | **ROA** |
| (A). Women on Board | | | (B). Female Executives | | |
| WoB | 0.1424 *** | 0.0237 *** | FE | 0.1587 *** | 0.0355 *** |
| t-Statistic | 10.3963 | 5.0314 | t-Statistic | 38.5677 | 8.1083 |
| Sales Growth | 0.0003 *** | 0.0002 *** | Sales Growth | 0.0003 *** | 0.0002 *** |
| t-Statistic | 4.5794 | 5.4255 | t-Statistic | 4.2632 | 5.3940 |
| Total Debt to Total Assets | −0.0204 | −0.1037 *** | Total Debt to Total Assets | −0.0162 | −0.1022 *** |
| t-Statistic | −0.5454 | −4.8326 | t-Statistic | −0.4271 | −4.6574 |
| C | 12.4273 *** | 7.8122 *** | C | 12.8713 *** | 7.7141 *** |
| t-Statistic R-squared | 10.3133 0.0103 | 11.8761 0.0510 | t-Statistic R-squared | 12.5413 0.0125 | 13.9565 0.0521 |
| Adjusted R-squared | 0.0098 | 0.0505 | Adjusted R-squared | 0.0120 | 0.0516 |
| F-statistic | 19.3396 | 101.9287 | F-statistic | 23.1081 | 102.8941 |
| Prob(F-statistic) | 0.0000 | 0.0000 | Prob(F-statistic) | 0.0000 | 0.0000 |
| Total observations | 9253 | 9482 | Total observations | 9131 | 9359 |
| Cross-sections included | 3189 | 3248 | Cross-sections included | 316 | 3219 |
| Periods Effects | Fixed | Fixed | Periods Effects | Fixed | Fixed |
| Cross-Section Effects | None | None | Cross-Section Effects | None | None |
| Ceof. Covariance method | White cross-section | White cross-section | Ceof. Covariance method | White cross-section | White cross-section |
| (b) | | | | | |
| **Dependent Variable** | **ROE** | **ROA** | **Dependent Variable** | **ROE** | **ROA** |
| (A). Women on Board | | | (B). Female Executives | | |
| WoB | 0.1620 *** | 0.0366 *** | FE | 0.1817 *** | 0.0530 *** |
| t-Statistic | 13.1178 | 9.6556 | t-Statistic | 58.4296 | 11.4235 |
| Sales Growth | 0.0003 *** | 0.0002 *** | Sales Growth | 0.0003 *** | 0.0002 *** |
| t-Statistic | 4.5020 | 5.2494 | t-Statistic | 4.1675 | 5.2092 |
| Total Debt to Total Assets | −0.0468 | −0.1323 *** | Total Debt to Total Assets | −0.0412 | −0.1310 *** |
| t-Statistic | −0.9768 | −5.3840 | t-Statistic | −0.8443 | −5.1741 |
| C | 13.0048 *** | 8.9963 *** | C | 13.5065 *** | 8.9292 *** |
| t-Statistic R-squared | 8.9592 0.0115 | 12.0761 0.0679 | t-Statistic R-squared | 10.1500 0.0143 | 13.7158 0.0706 |
| Adjusted R-squared | 0.0109 | 0.0674 | Adjusted R-squared | 0.0136 | 0.0700 |
| F-statistic | 17.9808 | 115.74230 | F-statistic | 21.9988 | 118.8701 |

**Table 4.** *Cont.*

| (b) | | | | | |
|---|---|---|---|---|---|
| **Dependent Variable** | **ROE** | **ROA** | **Dependent Variable** | **ROE** | **ROA** |
| Prob(F-statistic) | 0.0000 | 0.0000 | Prob(F-statistic) | 0.0000 | 0.0000 |
| Total observations | 7721 | 7944 | Total observations | 7602 | 7825 |
| Cross-sections included | 2667 | 2724 | Cross-sections included | 2639 | 2825 |
| Periods Effects | Fixed | Fixed | Periods Effects | Fixed | Fixed |
| Cross-Section Effects | None | None | Cross-Section Effects | None | None |
| Ceof. Covariance method | White cross-section | White cross-section | Ceof. Covariance method | White cross-section | White cross-section |

*** 1% statistic significant.

The results were considered very solid, as the estimated coefficients did not change materially, regardless of the method, sample, and variables under consideration, while changing the sample (all sectors or excluded financials) and the independent variables, either using the percent of women on the board of directors or the percent of female executives.

The empirical models show that if the percent of women on board increases by 10%, then the profitability (ROE) will increase by 1.4% to 1.6%. Equivalent results can also be extracted by applying the percent of female executives (10% higher percent of female executives will increase the profitability ratio by 1.6% to 1.8%). Overall, the pandemic crisis does not seem to negatively affect either the tendency of firms to support and choose women in positions of responsibility or their positive correlation with the firms' performance.

## 5. Conclusions

Our research interest leads us to seek answers to questions related to the role of women in executive positions on corporate boards worldwide. The key question under investigation is whether the participation of women in corporate decisions can lead firms to profitable outcomes. For this purpose, there are used as dependent variables two different proxies (ROE and ROA) as indicators of corporate profitability measurement and two ESG variables in order to approach the role of women in taking corporate decisions. In addition to the participation of females on firms' profitability, this study uses sales growth and financial leverage as deterministic variables as key factors affecting firms' profitability.

Our sample is composed of listed companies worldwide, across all geographical regions and activity sectors, in order to acquire a clearer and more robust picture of the situation under shaping. Thus, we used both cross-sectional and panel regressions to move closer to our target.

A remarkable outcome of this study is that both the percent of female executives and the percent of women participating in boards of directors increased gradually in 2021 compared to the previous years, indicating that women have increased their influence in corporate management around the world. The results of this study are not differentiated regardless of the factors, and the model's specification is applied to the total sample of companies or to the sample excluding financial sector companies.

A very interesting and remarkable outcome of the empirical analysis is that when the percent of women on board increases by 10%, the index of profitability will increase by 1.4% to 1.8%, regardless of the measurement of female participation in executive positions used. Moreover, the conclusions derived from the impact of females holding executive positions on firms' performance do not seem to change over time. That led us to conclude that regardless of the conditions and the effect on firms from the external environment (i.e., the

prevalence of pandemics or not), the impact of female executives on firms' profitability is significant and material during the time horizon of our analysis.

Taking into mind that the statistically significant results of all the model's specifications and econometric methods verify the stability of the approach, it has to be considered that the years included in the sample are considered very volatile due to the impact of the 2020 pandemic crisis and the strong recovery of 2021. However, despite the high volatility in corporate data, the empirical results of our models were very stable.

Last but not least, as can be perceived from the above analysis, the study's findings can be considered of high importance for all stakeholders, each in light of his own interests and points of view. However, the current analysis could, at a later stage, face some specific emerging limitations. The first impediment is structural and cannot be easily confronted. More specifically, the data under consideration gives information about current conditions and cannot necessarily be a precursor of future economic activity. So far, the estimated sensitivities and empirical results of our study may have limited predictive ability. A second limitation of our study is the sample structure. For instance, for research purposes, a sample is used that includes both companies from developed and developing economies worldwide, without apportioning them according to geographical and sectoral criteria. However, it is worth mentioning that the authors intention is to further expand this study by examining the differences per state and geographical region, so as to specify considerable issues of mentality, cultural background, and how each society receives and corresponds to human rights generally and in labor markets. In addition to the above, the authors have in mind to proceed to an in-depth analysis by sector of activity, as it is very interesting to perceive the real situation and the recent developments in the so-called "male dominated" sectors.

However, despite the above limitations, this study could be considerably contributable to policymakers, shareholders, investors, and the academic community. More specifically, from an academic perspective, this study could provide new, interesting findings to the existing literature, as it differentiates the analysis by taking into consideration all economic activity sectors worldwide before and during the pandemic period and applying different proxies for measuring women's impact on firms' performance.

A main research interest that led us to proceed with this study is the official expressed position of the United Nations and EU related to the importance and upgraded role of women both in society and the labor market. Thus, the results of our study could be taken into consideration by firms, shareholders, institutions, and other stakeholders involved in the struggle to eliminate gender discrimination and upgrade female roles in the labor market.

The participation of women on the board of directors can contribute to the improvement of corporate governance by bringing gender diversity, diverse viewpoints, and experiences to board discussions. This can result in better decision-making and reduced risk-taking. Moreover, policies promoting the participation of women on the board of directors can help promote gender equality in society and the workplace. These policies can also help reduce discrimination against women in leadership positions. Additionally, as already stated, policies promoting increased participation of women in positions of high responsibility can also promote greater social responsibility, as there is evidence that diverse boards of directors tend to be more socially responsible and attentive to the community in which the company operates.

According to the preceding analysis, companies with greater gender diversity on the board of directors tend to have better financial performance. This will be to the benefit of shareholders and firms' managers, as increasing their trust in placing women in executive positions will increase their chances of achieving their goals related to their firms' financial performance. More specifically, this diversity could help firms tap into new markets, adapt to changing customer needs and preferences, and benefit from different perspectives and ideas. Last but not least, the participation of women in executive positions can help to develop the skills and competencies of women leaders, promoting their professional

advancement and increasing their representation in other fields, reinforcing economies' human capital.

Overall, promoting the participation of women on the board of directors can have positive policy implications for corporate performance and governance, social equality, competitiveness, and the development of human capital.

**Author Contributions:** Conceptualization, I.K.; Methodology, I.K. and K.A.; Software, K.A.; Validation, K.A.; Data curation, K.A.; Writing—original draft, C.B.; Writing—review & editing, C.B. and I.K.; Visualization, C.B. and I.K.; Supervision, I.K. All authors have read and agreed to the published version of the manuscript.

**Funding:** This research received no external funding.

**Institutional Review Board Statement:** Not applicable.

**Informed Consent Statement:** Not applicable.

**Data Availability Statement:** The data that support the findings of this study are available from the corresponding author upon reasonable request.

**Conflicts of Interest:** The authors declare no conflict of interest.

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
