# Peer review of "Women’s Participation in Firms’ Management and Their Impact on Financial Performance: Pre-COVID-19 and COVID-19 Period Evidence"

_sustainability, doi:10.3390/su15118686_

Round 1

Reviewer 1 Report

The methodology employed is very poorly presented, and the tables of results are quite misleading and messy. I advice a general reformulation of the results presentation. Moreover, also the variables used are scarcely justified. Be aware on the dataset construction and the model specification!!! 

Author Response

Comment 1:

The methodology employed is very poorly presented, and the tables of results are quite misleading and messy. I advice a general reformulation of the results presentation.

Answer:

The authors thank the reviewer for the comment.

We restructured data and methodology part and we added the hypothesis part in this section. Moreover, we proceeded to a more thorough analysis of the methodology applied, in order to be more comprehensive and we analyzed in detail all the specifications of the model applied.

Furthermore, we restructured the tables in order to be presented in a more clear way, so to be more comprehensive and we proceeded to specific amendments to the way of explaining and presenting the empirical results.

Comment 2:

Moreover, also the variables used are scarcely justified. Be aware on the dataset construction and the model specification!!! 

Answer:

The authors thank the reviewer for the comment.

We explained even more the variables of the model and the data set construction, and we addressed some specific literature related to the variables’ approach. 

Reviewer 2 Report

The article raises an important topic of corporate gender equality and female leadership. 

It could be interesting to make several improvements to the article:

1 - Please verify the citation style guide. Some in-text citations don't comply with the citation norms.

2 - What is the function of the footnote on page 6?

3 - Tables 2 and 3 mention each year four times. This is a bit confusing. It could be interesting to have a more precise time framework (for example Q1, Q2, Q3, Q4 of each year).

4 - The concluding section is way too short. It might be useful to either add 4-5 paragraphs with a more detailed summary of your research findings and their impact on the academic field under consideration or add the section "Discussion" right before the section "Conclusions".  

Author Response

Comment 1:

Please verify the citation style guide. Some in-text citations don't comply with the citation norms.

Answer:

The authors thank the reviewer for the comment.

We will check the citation style guide thoroughly at the proofreading stage, once the paper will be accepted for publication, following the official guidelines of the JEL.

Comment 2:

What is the function of the footnote on page 6?

Answer:

We removed the comment, as it was corresponding to the description of a classification process, which is unnecessary for the purposes of our analysis.

Comment 3:

Tables 2 and 3 mention each year four times. This is a bit confusing. It could be interesting to have a more precise time framework (for example Q1, Q2, Q3, Q4 of each year).

Answer:

The authors thank the reviewer for the comment.

We restructured the tables mentioned in order to be presented in a more clear way and to be more comprehensive.

More particularly, the years are addressed to accounting periods and are not divided into quarter periods. For each of three years of the study correspond two ways of determining firms’ performance (dependent variable); meaning ROE and ROA.

Comment 4:

The concluding section is way too short. It might be useful to either add 4-5 paragraphs with a more detailed summary of your research findings and their impact on the academic field under consideration or add the section "Discussion" right before the section "Conclusions".  

Answer:

The authors thank the reviewer for the comment.

We proceeded to restructure and main amendments in the conclusion section and we added limitations, thoughts for further research and we analyzed even more the policy implications in order to be more comprehensive.

Finally, we proceeded to the improvement of the English language writing.

Reviewer 3 Report

This paper present interesting topic from under research area. Is nice to see some papers from underdeveloped countries as they provide new perspective on how systems from developed countries operate in these countries. Having said this, I have some concerns and suggestion for improvement.

1-            The introduction should also include paragraph that summarize the findings of the study and another paragraph that provide contribution of the study based on the findings. Moreover, this section should involve the contribution of study.

2-            Literature Review section should be updated with recent ciation like 2022, 2023.

4-            the discussion of result must provide. So, why Bahrain and not any other country in the region that has similar characteristics (or is this country an individuality compared to others in the region)? Concluding at this point, the author(s) did not give consistent arguments to support the Bahrainempirical research in this field, which seriously affects the value of the study.

5-            The conclusion should include limitations of the study and suggestion for future research.

6-            In the conclusions, introduce recent updated bibliography, and comment on it in light of the literature. In this regard, please include the following works:

 2022). Do corporate governance and top management team diversity have a financial impact among financial sector? A further analysis. Cogent

Business & Management, 9(1), 2141093. (2022). the role of women on board of directors and firm performance: evidence from Saudi Arabia financial market. Corporate Governance and Organizational Behavior Review, 6(3), 44-55

7-            Finally, i am not native speaker but i suggest author/s to do professional proofreading.

Author Response

Comment 1:

The introduction should also include paragraph that summarize the findings of the study and another paragraph that provide contribution of the study based on the findings. Moreover, this section should involve the contribution of study.

Answer:

The authors thank the reviewer for the comment.

We added a paragraph that verifies and highlights the importance of our findings related to the role and importance and impact of female executives on firms’ performance. Moreover, there is a paragraph focusing on the contribution of our paper to shareholders, managers, academic research, society and labor market. Besides that, the contribution and policy implications are more thoroughly analyzed in the conclusions’ section. 

Comment 2:

Literature Review section should be updated with recent citation like 2022, 2023

Answer:

The authors thank the reviewer for the comment.

We proceeded to restructure of the literature review section and we updated it, adding more current bibliography.

Comment 3:

The discussion of result must provide. So, why Bahrain and not any other country in the region that has similar characteristics (or is this country an individuality compared to others in the region)? Concluding at this point, the author(s) did not give consistent arguments to support the Bahrain empirical research in this field, which seriously affects the value of the study.

Answer:

The authors thank the reviewer for the comment.

We didn’t exactly understand this comment. However, we proceeded to specific amendments and restructure of the data and methodology section, we modified the tables and we rewrote our results in way to be more clearly presented and comprehensive. 

Comment 4:

The conclusion should include limitations of the study and suggestion for future research.

Answer:

The authors thank the reviewer for the comment.

We proceeded to restructure and main amendments in the conclusion section and we added limitations, thoughts for further research and we analyzed even more the policy implications in order to be more comprehensive.

Comment 5:

In the conclusions, introduce recent updated bibliography, and comment on it in light of the literature. In this regard, please include the following works:

 2022). Do corporate governance and top management team diversity have a financial impact among financial sector? A further analysis. Cogent

Business & Management, 9(1), 2141093. (2022). the role of women on board of directors and firm performance: evidence from Saudi Arabia financial market. Corporate Governance and Organizational Behavior Review, 6(3), 44-55

Answer:

The authors thank the reviewer for the comment.

As stated above, we updated the literature review section with recent and appropriate bibliography.

Comment 6:

Finally, I am not native speaker but i suggest author/s to do professional proofreading.

Answer:

The authors thank the reviewer for the comment.

We further improved the written English through proofreading.

Reviewer 4 Report

The abstract must be structured: Purpose: (text...); Results and contributions: (text...); Methodology: (text...); Gap: (text...); Relevance: (text...); Impact: (text...). Also the summary is too long. It should not exceed 150 words.

Significant of the paper:

This paper contains sufficiently knowledge. The purpose of the investigation is relevant.

Literature:

The paper is well related to previous literature. The author demonstrates extensive knowledge of previous studies. The author should do an additional task of searching for recent work and incorporating innovative arguments in his research

Methodology: 

The research method is appropriate and is built on the basis of the previous work through surveys.

Results:

The research results are presented clearly and are similar to those obtained in the previous literature. However, it is necessary to increase the sample period.

Implications for research:  

The practical and social implications are very relevant. However, it is important that the authors are more incisive in the conclusions drawn.

Author Response

Comment 1:

The abstract must be structured: Purpose: (text...); Results and contributions: (text...); Methodology: (text...); Gap: (text...); Relevance: (text...); Impact: (text...). Also the summary is too long. It should not exceed 150 words.

Answer:

The authors thank the reviewer for the comment.

We proceeded to the abstract’s restructure in order to meet the comment.

Comment 2:

The paper is well related to previous literature. The author demonstrates extensive knowledge of previous studies. The author should do an additional task of searching for recent work and incorporating innovative arguments in his research. 

Answer:

The authors thank the reviewer for the comment.

We proceeded to restructure of the literature review section and we updated it, adding more current bibliography.

Comment 3:

The research results are presented clearly and are similar to those obtained in the previous literature. However, it is necessary to increase the sample period.

Answer:

The authors thank the reviewer for the comment.

At the first paragraph of “Data and Methodology” section we analyzed succinctly the reasons of including this specific period for our research process.

More specifically, the final year of the current research is 2021, as it is the last year of availability of official data registered. 2019 is the starting year of the conducted research in order to include the corporate results of the year preceding the break out of the pandemic shock. Under the same rationale, 2020 corresponds to the effects of Covid-19 and year 2021 was applied to evaluate the post covid-19 consequences.

 Comment 4:

The practical and social implications are very relevant. However, it is important that the authors are more incisive in the conclusions drawn.

Answer:

The authors thank the reviewer for the comment.

We proceeded to restructure and main amendments in the conclusion section and we added limitations, thoughts for further research and we analyzed even more the policy implications in order to be more comprehensive.

Round 2

Reviewer 4 Report

I consider that the paper satisfies the publication conditions, after the changes made.